# Considerations for temperature sensor placement on rotary-wing unmanned aircraft systems

Brian R. Greene[1,2,3], Antonio R. Segales[2,3,4], Sean Waugh[5], Simon Duthoit[2,3,4], and Phillip B. Chilson[1,2,3]

[1]University of Oklahoma School of Meteorology, Norman, Oklahoma
[2]Advanced Radar Research Center, University of Oklahoma, Norman, Oklahoma
[3]Center for Autonomous Sensing and Sampling, University of Oklahoma, Norman, Oklahoma
[4]University of Oklahoma School of Electrical and Computer Engineering, Norman, Oklahoma
[5]NOAA/OAR National Severe Storms Laboratory, Norman, Oklahoma

**Correspondence:** B. Greene (brian.greene@ou.edu)

**Abstract.** Integrating sensors with a rotary-wing unmanned aircraft system (rwUAS) can introduce several sources of biases and uncertainties if not properly accounted for. To maximize the potential for rwUAS to provide reliable observations, it is imperative to have an understanding of their strengths and limitations under varying environmental conditions. This study focuses on the quality of measurements relative to sensor locations on board rwUAS. Typically, thermistors require aspiration and proper siting free of heat sources to make representative measurements of the atmosphere. In an effort to characterize ideal locations for sensor placement, a series of experiments were conducted in the homogeneous environment of an indoor chamber with a pedestal-mounted rwUAS. A suite of thermistors along with a wind probe were mounted inside of a solar shield, which was affixed to a linear actuator arm. The actuator arm was configured such that the sensors within the solar shield would travel underneath the platform into and out of the propeller wash. The actuator arm was displaced horizontally underneath the platform while the motors were throttled to 50 percent, yielding a time series of temperature and wind speed which could be compared to temperatures being collected in the ambient environment. Results indicate that temperatures may be biased on the order of 0.5–1.0°C and vary appreciably without aspiration, sensors placed close to the tips of the rotors may experience biases due to frictional and compressional heating as a result of turbulent fluctuations, and sensors in proximity to motors may experience biases approaching 1°C. From these trials, it has been determined that sensor placement underneath a propeller on an rwUAS a distance of one quarter the length of the propeller from the tip is most likely to be minimally impacted from influences of turbulence and motor, compressional, and frictional heating while still maintaining adequate airflow. When opting to use rotor wash as a means for sensor aspiration, the user must be cognizant of these potential sources of platform-induced heating when determining sensor location.

# 1 Introduction

The planetary boundary layer (PBL) is the lowest layer of the troposphere which exchanges energy with the Earth's surface on timescales of less than one hour (Stull, 1988), and acquiring atmospheric measurements in this region has proven to be challenging (National Research Council, 2009; Hardesty and Hoff, 2012). PBL flows are highly complex and nonlinear in space and time, even with several layers of assumptions applied in theory. As such, it has always been a challenge for atmospheric scientists to collect representative measurements of the environment, even with continual advances in technology. One of the most common resources for PBL studies has been instrumented towers, which can continually provide data at a point location over long periods of time (e.g., Charba, 1974; Shapiro, 1984; Poulos et al., 2002). While highly reliable and configurable, instrumented towers do come with an inherent downside. Being limited in vertical extent, the convective boundary layer often extends well above even the tallest of towers. Even networks with the 30 km average horizontal resolution of the Oklahoma Mesonet (Brock et al., 1995; McPherson et al., 2007) still cannot provide details on the vertical structure of the atmosphere.

Presently, weather balloons are the most common tool available for in-situ observations above the level of towers. They provide valuable kinematic and thermodynamic data from the upper atmosphere, which impacts both short-term weather forecasts (Cohen et al., 2007; Faccani et al., 2009; Lackmann, 2011) as well as climatological trends (Luers and Eskridge, 1998; Lanzante et al., 2003; Thompson and Solomon, 2005) and can serve as a baseline for model verification (Agustí-Panareda et al., 2010; Benjamin et al., 2010; Gensini et al., 2014). Rawinsondes are launched in hundreds of locations around the world every day, although usually only twice a day at most sites. This operational network is also not suited to provide adequate PBL measurements, as they ascend too rapidly through the lowest levels (National Research Council, 2009). More frequent deployments with slower ascents are commonly performed on field campaigns (e.g., Kosiba et al., 2013; Parker, 2014; Trapp et al., 2016; Geerts et al., 2017), but this becomes expensive as the sensor package is rarely recovered for reuse. Specialized satellite remote sensors can derive vertical thermodynamic and kinematic profiles across significant areas of the Earth, but vertical resolutions in the PBL are too coarse for practical application.

Surface-based remote sensors such as wind profilers, Doppler lidars, sodars, and radiometers are capable of continuously observing a fixed location (e.g., Grund et al., 2001; Poulos et al., 2002; Banta et al., 2015; Bonin et al., 2015; Lundquist et al., 2017; Toms et al., 2017; Geerts et al., 2017; Blumberg et al., 2017), but rely on numerous assumptions about the atmosphere and have trouble resolving measurements close to the surface. These types of instruments are also cost-prohibitive when considering expansion to larger-scale networks such as the global upper-air sites.

Even when combining surface towers, balloons, and remote sensors with other observational techniques such as tethered balloons, Doppler weather radars, and satellite remote sensors, the National Research Council (2009) still concluded that the "vertical component of U.S. mesoscale observations is inadequate." The NRC in this report implored government agencies to pursue developments in capabilities to monitor the lower atmosphere at finer scales in space and time.

Capitalizing on the recent commercial accessibility of small unmanned aircraft systems (sUASs) and miniaturized sensor packages, numerous groups around the world have embraced the potential for integrated platforms to fill this atmospheric data void (e.g., Reuder et al., 2009; Houston et al., 2012; Lothon et al., 2014; Wildmann et al., 2014; Båserud et al., 2016; de Boer

et al., 2016; Bailey et al., 2017; Vömel et al., 2018; Koch et al., 2018). UASs have the notable advantage of being able to operate in regions beyond the reach of typical systems, including environments that may be dangerous. Sophisticated systems can be deployed on a regular basis for consistent measurements, and they are less expensive alternatives to ground-based remote sensors measuring similar parameters. They can be used for a variety of missions measuring different quantities, such as horizontal transects across airmass boundaries or continuous vertical profiling at fixed ascent rates. Owing in part to the longstanding history of manned research aircraft, fixed-wing UASs (fwUASs) have been at the forefront of UAS development for atmospheric research (Saïd et al., 2005; Gioli et al., 2006; van den Kroonenberg et al., 2012). However, fwUASs come with several notable disadvantages, namely their inability to sample a vertical column at a fixed horizontal position, risks when operating close to the ground, and the need for a suitable surface for landing and possibly takeoff. Currently, rotary-wing UASs (rwUASs) are being proven to be a viable supplement to fwUASs thanks to their autonomous vertical takeoff/landing capabilities (Brosy et al., 2017; Vömel et al., 2018). Integration of rwUAS into observational networks and research efforts has the potential to vastly improve our understanding of processes occurring in the lowest regions of the atmosphere at unprecedented scales. It is therefore imperative for data collected by these platforms to achieve the highest possible degree of environmental representativeness as rwUAS become more commercially accessible.

As one application, rotary-wing UAS can be used to measure the same thermodynamic and kinematic properties as radiosondes: pressure, temperature, relative humidity, and horizontal wind speed and direction (Neumann and Bartholmai, 2015; Brosy et al., 2017). However, when making measurements, the platform itself can influence these observations. There are a number of factors that can affect any single measurement and each observation must be carefully designed and examined to ensure that it is as free as possible from external influence. In this study, we focus on the effects of sensor placement on temperature observations.

To ensure that a thermometer, such as a thermistor, produces accurate measurements, it is critical that the sensor be shielded from solar radiation and properly aspirated with the ambient environment (Tanner et al., 1996; Richardson et al., 1999; Hubbard et al., 2004). Moreover, sensor self-heating can lead to significant measurement bias in some thermistors if not properly accounted for. Thermistors use a temperature-sensitive resistor to measure temperature. By knowing the input voltage and measuring how it changes across the thermistor, the resistance of the temperature sensitive resistor can be determined, and thus the air temperature. If current is run constantly through the resistor, heat is generated. Such a sensor must be properly aspirated or the resultant heat can modify the ambient environment, thereby influencing the measurement itself.

Observations of temperature from tower-mounted thermistors typically utilize solar-shielded chambers with fans to mechanically aspirate the sensors to improve data quality (Brock et al., 1995; McPherson et al., 2007). However, when considering the integration of sensors into a rwUAS, utilizing a fan poses a dilemma. Although a fan could ensure proper aspiration, it would draw power and add weight to the platform and potentially significantly decrease flight duration. Therefore, an alternative solution to this problem is to aspirate the sensors with the air currents produced by the rotating propellers (i.e., rotor wash). However, due to the complex flow around a rwUAS in flight, the location on the rwUAS providing the best aspiration is not obvious. If exposed to too little airflow, the sensor could self heat or not adequately sample the ambient atmosphere. If exposed to too much airflow, compressional heating of the air stream becomes an issue (Rodert, 1941). Furthermore, heat from the ro-

tary motor can also alter the measured air. Flow in the proximity to the propeller tips is also associated with the highest values of turbulent intensity and temperature fluctuations (Swean and Schetz, 1979). After initial experiments involving rwUAS for research efforts, it was determined that a more in-depth examination of sensor location was needed to ensure data quality.

## 2 Equipment

With the questions surrounding temperature sensor placement on a rwUAS, an experiment was created to objectively determine the optimal location for quality temperature measurements, which is the primary focus of this study. A summary of the findings from the experiment is discussed below.

### 2.1 Rotary-wing aircraft

The University of Oklahoma's custom-built CopterSonde rwUAS (Figure 1) facilitates a symmetrical carbon fiber hashtag
design with a diameter of 65 cm and is driven by eight brushless electric motors and 25 cm diameter propellers. The maximum payload mass amounts to 1 kg with a total all-up weight of about 7 kg. The maximum flight time is about 20 minutes. The CopterSonde has a top flight speed of 25 m s$^{-1}$ and thus can be flown safely in winds up to a maximum horizontal speed of 20 m s$^{-1}$. The CopterSonde is equipped with a Pixhawk autopilot (3D Robotics, Inc.) which relies on an on-board inertial measurement unit (IMU) for attitude estimation. A barometric pressure sensor is used to control the altitude of the rwUAS. It
also carries a post-processing kinematic differential GPS unit which gives centimeter positioning accuracy in space. External sensor data are sent to the Pixhawk via the I$^2$C protocol, which are processed in parallel to the flight controls. This setup allows for a single consolidated data stream sent to the ground station over wireless radio using the Mavlink protocol. The operative distance of the communication system is around 5 km, capable of 15 km with upgraded antennas.

### 2.2 Temperature sensors

The CopterSonde platform utilized PT 100 thermistors distributed by International Met Systems (iMet) to make temperature observations. These bead thermistors offer a response time of 2 seconds in still air (approximately 1 second with 5 m s$^{-1}$ aspiration) over a range of -95 to +50°C, with an accuracy of $\pm$ 0.3°C and a resolution of 0.01°C. They are similar to the kind of sensors used on many standard radiosondes and are ideal for use on a rwUAS. The sensors were validated using an Oklahoma Climatological Survey aspirated chamber located outside the National Weather Center in Norman, Oklahoma. Offsets for each
sensor were determined over a period of several hours of a typical late afternoon in early spring, with temperatures comparable to room temperature. Thus, these offsets provide an implicit self-heating correction while aspirated.

The bead thermistors are one component of a bigger iMet system specifically designed for UAV applications: the iMet-XF UAV. This system uses a main board to which different types of thermodynamical sensors can be interfaced. It samples each connected sensor successively, including on-board GPS and pressure sensors, and provides these data in packets through serial
communication. Acquiring and storing the data can be achieved in a variety of ways: using a pair of radios to stream data to a ground station, or the unit can be connected to a computer for direct data streaming using the provided iMet software.

On the CopterSonde, the iMet thermistors were utilized in conjunction with a custom data acquisition unit. These data were then streamed and recorded on a ground station computer using a radio frequency link. The iMet temperature sensors are equipped with an integrated circuit board that converts the analog data stream into an $I^2C$ format. To streamline data input to the CopterSonde's flight controller (Pixhawk), a custom circuit board was developed that is capable of accepting and synchronizing 8 separate $I^2C$ sensor inputs and converting them to a single output data stream. A module was programmed for the Pixhawk to sample each sensor successively at a given rate, log their data on-board, and stream live data to the ground station.

## 2.3   NSSL Mobile Mesonet

To provide a comparison with the temperature data recorded using the iMet thermistors, a modified version of the National Severe Storms Laboratory (NSSL) Mobile Mesonet (MM) rack was used. The equipment rack, normally mounted to the roof of a vehicle, is capable of temperature, pressure, wind speed and direction, relative humidity, and solar radiation observations. For the tests presented here, the equipment rack was mounted to a cart. This allowed the rack to be placed in close proximity to the CopterSonde during measurements.

To measure temperature, two Campbell Scientific model 109 thermistors (CS 109) were used. One was mounted inside an aspirated radiation shield and one was mounted to the CopterSonde. The CS 109 has less than $0.03°C$ linearization error over the range of -50 to $+70°C$, with a $\pm 0.2°C$ tolerance between 0 and $70°C$. The thermistor is contained within a stainless steel housing, meant for use in damp conditions such as soil moisture measurements, and as such has a response time of 30 s with 5 m s$^{-1}$ airflow. The sensors used for this study had been recently calibrated in the Oklahoma Mesonet calibration lab to ensure its accuracy.

## 2.4   Oklahoma Mesonet hot-wire anemometer

In addition to measuring temperature, it is also relevant to measure the flow rate at the temperature sensor location. Doing so can indicate the extent to which the probes are in a "well mixed" environment and provides insight to the conditions that are being experienced by the sensors themselves. A Thermo Systems Inc. (TSI) hot-wire anemometer was used to gather precise velocity measurements about 1 cm from the temperature sensor mounting locations. On these scales, special considerations regarding the anemometer as a heat source were also required. As will be discussed in Section 4, a separate trial to control for possible interference was also conducted.

## 2.5   Linear actuator arm

The overall goal of the experiment was to find locations on a rwUAS where temperature readings are most representative of the environment. With this in mind, data collected at multiple locations on the rwUAS were examined to determine where the sensors experience bias relative to ambient air. To achieve this, the thermistors were placed on a linear actuator arm capable of moving the sensors horizontally directly underneath two of the motor mounts (Figure 2). The initial starting position at point A was 6.5 cm outside of one propeller, and the the sensor position was systematically stepped 0.24 cm per increment with a

dwelling time of two seconds per increment across the width of the CopterSonde towards point I. This motion of the arm was controlled by the same procedure on the ground station communicating with the hot-wire anemometer, which then recorded arm position, wind speeds, and computer timestamps at each step. The ending location was 12.5 cm outside the opposite side of the rwUAS, and took approximately 35 minutes to complete the process. Two of the iMet temperature sensors were attached to the arm, as was a CS 109 temperature sensor from the NSSL MM rack. Combining these three different datasets (NSSL logs for the NSSL probes, Pixhawk logs for the temperature sensors, and the computer logs for the arm and the anemometer) with their common timestamps therefore allowed for synchronized analysis of the separate data streams.

## 3   Thermistor self-heating experiment

Prior to analysis of the sensor placements (detailed in Sections 4 and 5), a baseline characterization of the iMet thermistor self-heating is required. A simple experiment consisting of three thermistors and a ducted fan was conducted to isolate the effects of aspiration. The fan was located at the base of a solar shield duct, which was bent at a right angle with sensors inserted through holes along the top. With the fan switched on, this configuration induced airflow to enter from the horizontal, pass across the sensors, and exhaust downwards through the fan.

The sensors were initially powered on to collect data while the fan was off. After holding this condition for approximately six minutes, the fan was powered on, pulling air across the sensors at 6 m s$^{-1}$. The fan was then alternately switched on and off for periods of two to three minutes each, for a total of three cycles.

Analysis of the iMet temperature responses (Figure 3) revealed that the sensors closely tracked with one another while the fan was on, with only a linear offset. It is immediately apparent that the sensors react to airflow through the solar shield, as temperatures repeatedly drop over 1°C in under 20 s. The temperature increase observed in all three sensors around minute 8 is due to a researcher stepping in front of the setup to take a photograph. While small, the aspirated sensors were capable of picking up on the influence of body heat and respond in a similar fashion. Prior to the first time the fan was switched on, the setup was idle for several minutes. However, following the fan being switched off, the sensors indicated temperature increases. For these periods, the observed heating was likely due to a combination of both sensor-self heating and the fan motor radiating heat upwards towards the sensors. This hypothesis is supported by iMet sensor 2 being in closest proximity to and directly above the fan while also heating the most rapidly. Furthermore, the other two sensors were higher up and displaced horizontally due to the geometry of the duct, and they showed slower heating rates while the fan was off. The key evidence for sensor self-heating is that the sensors return to their same respective temperature levels each time the fan was on, regardless of how they behaved while the fan was off. Therefore, this supports the requirement of sensor aspiration to properly measure the environmental temperature.

## 4 Anechoic chamber experiments

### 4.1 Setup

The University of Oklahoma Radar Innovations Laboratory has a large anechoic chamber used for calibration and testing of radar components and other electronic equipment. This chamber, however, also provides a reasonably homogeneous environment for testing when it is necessary to isolate the effects of various sensor influences on a rwUAS without solar radiation concerns or changes to the ambient environment. To offset the vertical variations in temperature that could exist in such a room, a common carpet fan was aimed at 45° from horizontal and turned on to maximum airflow about 15 minutes before the experiment to mix the environmental air. The CopterSonde was mounted on a large pedestal near the center of the room with a bracket that accommodated the vehicle and a linear actuator arm as previously mentioned (Figure 4A).

To simulate the wind flow of the aircraft in flight, the iMet and CS 109 thermistors and hot-wire anemometer were positioned inside of a 3D-printed plastic solar shield (Figure 4B, C). Due to the spatial constraints of the setup, the CS 109 was mounted vertically and underneath the iMet and wind sensors in an effort to measure the same air stream. Furthermore, to avoid bias in temperature measurements, the hot-wire anemometer was removed for the final round of testing. For each experiment, the actuator arm was mounted so that the sensors would pass directly underneath the motor mounts as the linear actuator arm moved horizontally (Figure 2).

To provide a reference temperature of the ambient environment, a second CS 109 sensor on the NSSL MM rack was mounted inside the aspirated U-tube radiation shield (not depicted, see Waugh and Fredrickson, 2010). A second iMet thermistor was also suspended 50 cm below the CopterSonde, allowing for reasonable (but turbulent) aspiration, as determined from previous trials not included in this study. The additional measurements provided by the suspended iMet thermistor and the CS 109 probe inside the radiation shield, were used to measure the "ambient" environment. For the purposes of these tests, the autopilot inputs to the motor throttle were bypassed, allowing for direct manipulation of throttle input using an external device.

### 4.2 Procedure

To begin, power was supplied to the iMet and NSSL thermistors and they began logging data. For the first trial, the motor position began at point A (6.5 cm horizontally from the tip of the nearest propeller, Figure 2), and the battery was connected with throttle at zero, allowing the sensors to sample an unaspirated environment for 8 minutes. After this period, the CopterSonde was throttled up to approximately 55 percent maximum power to simulate the airflow typical during slow ascent. Although not directly under the propellers, airflow across the sensors was 2 m s$^{-1}$, sufficient enough for aspiration. This position was sustained for 2 minutes before powering the motors off again. The sensors then remained in quiescent conditions for 2.5 minutes before throttling to 55 percent again. After giving the sensors 40 seconds to aspirate, the linear actuator was then incrementally moved towards point B by 0.24 centimeters, holding each position for 2 seconds. In total, it moved approximately 71.1 cm, which was outside of the rotor wash on the "I" side of the configuration. This experiment took a total of approximately 35 minutes to complete.

To control for the effects of potential heat advection from the hot wire anemometer, the same test was conducted after removing it from the solar shield. During this second trial, however, the initial start-stop-start of the motors was not performed. The CopterSonde and sensors were powered on for 2.7 minutes, then the throttle was increased to 55 percent for 35 seconds before incrementing the linear actuator arm's position.

## 5 Results and discussion

### 5.1 Experiment 1 - wind probe in tube

In the first experiment, to account for the longer response function of the CS 109 probes and to make more appropriate comparisons, a moving 10 s average of the iMet temperature data was calculated before each analysis point (Figure 5). The wind speeds presented are the raw outputs. Furthermore, the hot-wire anemometer had not been calibrated prior to this experiment, and thus values displayed may not be absolute. Confidence in relative precision is still high, however.

The air flow velocity peaked near 17 m s$^{-1}$ before decreasing to near zero directly underneath the motor which clearly identified passage through the rotor wash of the propellers as the linear actuator moves from one side of the rwUAS to the other (points B, D, Figure 2). A second minimum was encountered between the two propellers, before a similar pattern was observed while the sensors passed under the second propeller. A gradual temperature increase of 0.5° was observed by both background temperature sensors over the course of the 35 min experiment, likely attributable to the mechanical mixing of the chamber environment.

This velocity pattern and associated temperature bias demonstrates that when considering sensor location for adequate airflow, directly under the motors or between the two propellers is not a viable option. While the first conclusion might be obvious, a relative minimum in the flow velocity was not expected between the propellers. In addition, differences do exist between the various sensors, and a steady increase in temperature on all sensors was measured over the duration of the experiment (Figure 5). To account for this steady increase, temperatures relative to the background environment are considered for the remainder of this discussion.

A closer look at the first 16 minutes of this analysis relative to the background temperature (Figure 6) reveals evidence of the self-heating phenomenon. For over 8 minutes, the probes in the solar shield recorded 0.2–0.4°C above the relatively constant background, with variations owing to the presence of the hot-wire anemometer. During this period, the motors of the CopterSonde were not on, thus no aspiration to the sensor existed. Once the motors initially throttled up (green dashed line), temperatures dropped to within 0.1°C of the reference, and remained in this range until the motors were shut off again 2 minutes later. Immediately after throttling down, temperature began rising again, by 0.5°C in under 3 minutes. Finally, when throttled back up again at the 13 minute mark, temperatures returned to anomalies of 0.1°C in under 30 seconds.

Although influences from the anemometer are likely inherent during this initial period, the overall response of the sensors to aspiration matches results from the experiment discussed in Section 3. It can therefore be concluded that rotor wash is capable of mitigating the decoupling of sensors from the ambient environment, so long as the sensors are free from other external sources of heating which will be discussed below.

After the motors turned on at the 13-minute mark, the actuator arm began translating underneath the aircraft. Due to the complexity of the flow field underneath propellers rotating at several thousand revolutions per minute, it is reasonable to believe that small nuances in temperature depicted can be caused by limits in sensor accuracy and sampling rates in turbulent flow. There are, however, several identifiable trends that are attributable to artificial sources such as motor heat and sensor decoupling (Figure 7).

At minute 15.5 (just prior to point B), the probes intercepted a warm stream of air likely owing to turbulent fluctuations and compressional heating on the tip of the propeller spreading down and outward along the periphery of the propeller wash. A similar observation is made on the other end of the CopterSonde at minute 28 (after point H). At minute 16 (point B), the sensors moved under the propellers and out of the warm air stream from the tips, allowing temperatures to stabilize within 0.2°C of the reference temperature. This pattern is consistently observed underneath the four peaks in wind speed, representing: exterior propeller 1 (point B), interior propeller 1 (point D), interior propeller 2 (point F), and exterior propeller 2 (point H), in order.

As the actuator arm moved the sensors underneath the CopterSonde's motor mounts from minutes 18–19.5 (point C) and 25–26.5 (point G), temperatures rapidly rose 0.7–1.0°C relative to the background over the course of 1 minute (Figure 7). Since temperatures began rising with wind speeds well above levels at the beginning of the experiment (2–4 m s$^{-1}$), the source of this increase was not necessarily solely due to self heating. Instead, their proximity to the motors leads to the conclusion that the sensors were intercepting hot air advected from the motors.

Finally, the sensors mounted on the arm sampled the space in between the interior propeller tips at minutes 21.5–23 (point E). At that time, wind speeds dropped to less than 3 m s$^{-1}$, similar to those at the initial position of the sensors. Subsequently, a small (0.1°C) temperature rise is noted in both the iMet and NSSL sensors. Because the aspiration rates were similar to those at the beginning of this experiment, the primary driver of this temperature rise was likely a combination of self-heating and intercepting a warm air stream originating from the hot-wire anemometer. In order to account for the influence of the hot-wire anemometer, a second analogous experiment was performed by removing it.

## 5.2 Experiment 2 - no wind probe

In Experiment 2, sensors were allowed to remain unaspirated for about 2.75 min before throttling up and moving the linear actuator arm, similar to Experiment 1 except without the initial aspiration test. Although the wind probe was removed, the actuator arm increments were identical, so it is reasonable to compare the temperature time series against the wind speeds from Experiment 1 (Figure 8). In general, the temperature pattern was largely similar to the results from Experiment 1: symmetrical about the center of the two propellers, increases in temperature on the outside tips (before point B and after point H), and large increases underneath the motor mounts (points C and G). However, the small rise in temperature in between both propellers (point E) was no longer observed, and the overall increase in temperature underneath the motor mounts was 0.2–0.3°C less than in Experiment 1. Therefore, the hot-wire anemometer likely biased temperature readings in this region of relatively stagnant flow. Finally, the CopterSonde's battery rapidly approached its critical level as the motors were shut off at min 18.5, so temperature trends after this mark should not be strongly considered.

## 6 Conclusions

One must take special consideration regarding sensor placement when attempting to measure environmental temperature using a rotary-wing UAS because these platforms are prone to modifying their own environments. In order to determine sensor locations free of systematic biases, multiple experiments were conducted in a relatively homogeneous chamber, where sensors were sequentially displaced underneath the rotor wash from a mounted rwUAS. Results from the two experiments presented provide useful guidelines with regards to sensor placement. Several sources of temperature bias exist, including (but not limited to) those observed in this study: sensor self-heating, compressional heating and turbulent fluctuations from the propeller-modified air streams, and heating from the motors.

In addition to the two experiments discussed previously, several test trials were also conducted. These additional experiments were almost identical to the setup described in Section 4, but were missing some key elements. For instance, the sensors were mounted to the linear actuator arm without a solar radiation shield. As a result, the sensors intercepted larger areas of influence from heat sources such as the motors and propeller tips when compared to sensors inside a shield. Furthermore, the environmental air was not mixed using a carpet fan prior to running the experiment. There was a much more noticeable increase in the chamber's background temperature during these trials, which reduced confidence in analysis and was difficult to reproduce. These tests ultimately still eluded to similar results in temperature sensor placement, prompting the more refined experiments as the focus of this study.

Given these results, out of the locations tested the optimal position for measuring environmental temperatures while hovering or ascending with a rotary-wing UAS is in a solar shield about 5–10 cm below the propeller and one third the length of the propeller from the tip. This location provides ample aspiration while avoiding the warm air streams from the motor and propeller tips. Other locations above or below the UAS run the risk of encountering stagnation in flow, which can exaggerate the effects of self-heating and generally decouple the sensor from the environment. Furthermore, proximity to external heat sources such as batteries or the rotary motors are also capable of introducing artificially warmed air streams. By following these general guidelines, it is of the authors' opinions that rwUAS are capable of obtaining trustworthy atmospheric measurements across a variety of applications.

*Code and data availability.* Data and code are available upon request to the corresponding author.

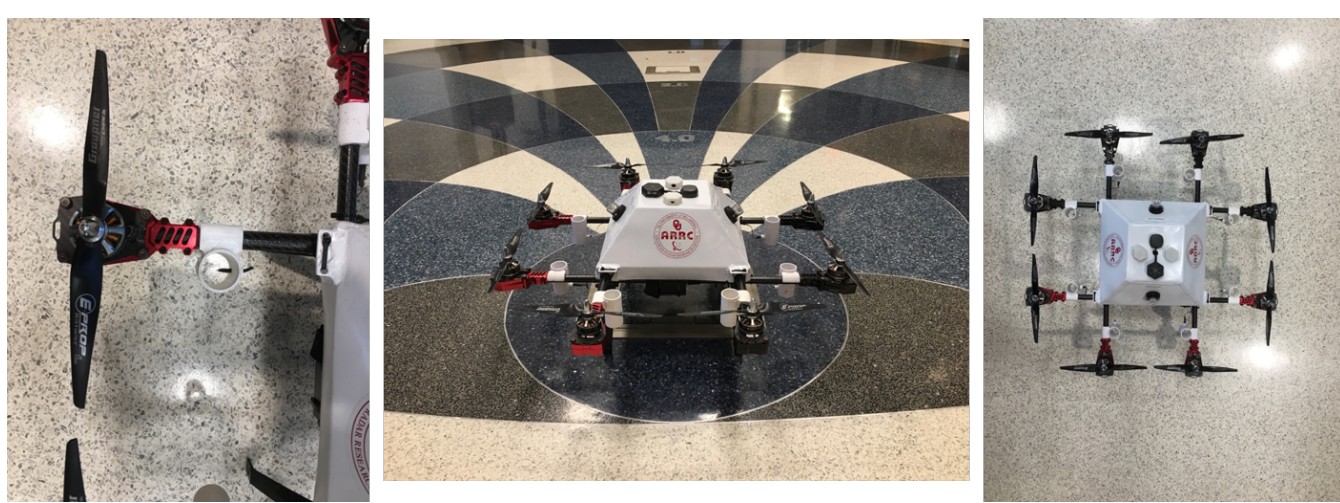

**Figure 1.** The CopterSonde, an octo-rotor UAS designed and built by the Center for Autonomous Sensing and Sampling at the University of Oklahoma.

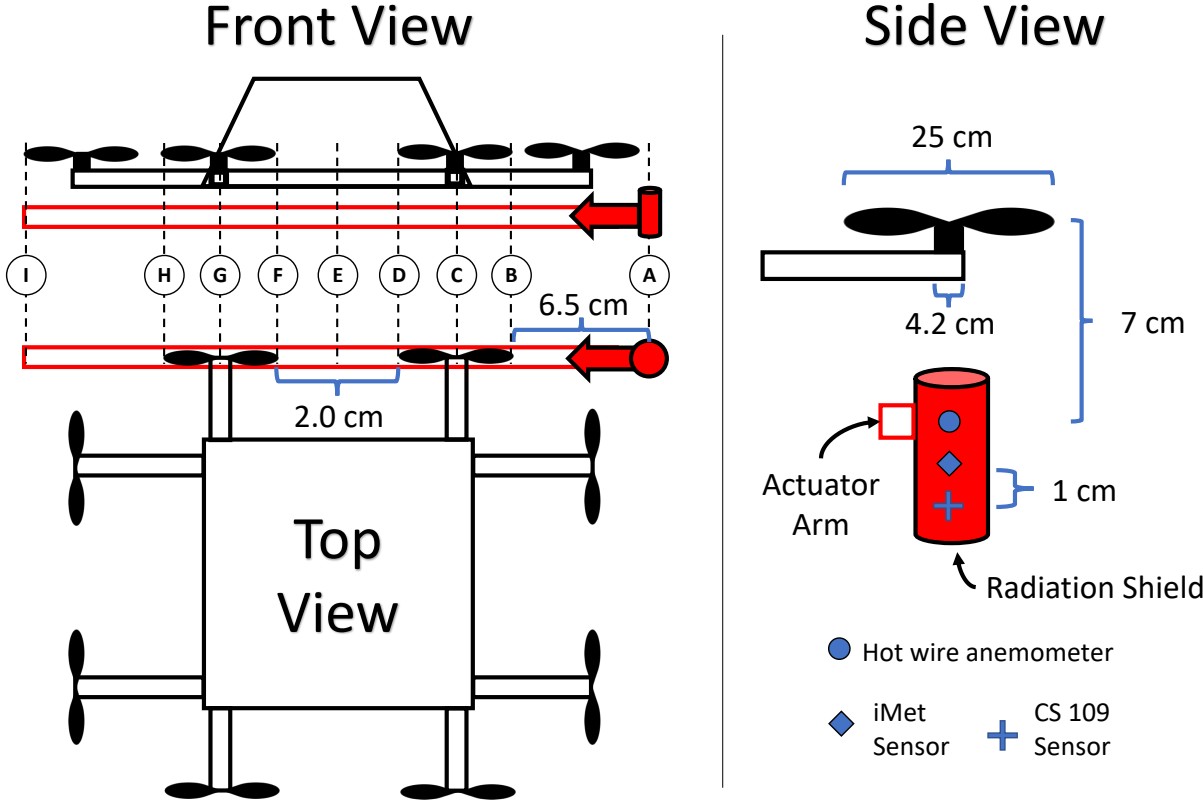

**Figure 2.** Schematic and dimensions of the rwUAS used in this study (drawing not to scale). In front and top view, the linear actuator arm is represented by the red rectangle outline, and the sensor package as a red circle. The arm was displaced from point A to point I, directly underneath the motor mounts and one pair of propellers as seen in the top-down and side views. Point B represents the tip of propeller 1, point C is directly under motor 1, D is the other side of the same propeller. Point E is halfway between the two propellers, and points F–H are symmetrical to points B–D.

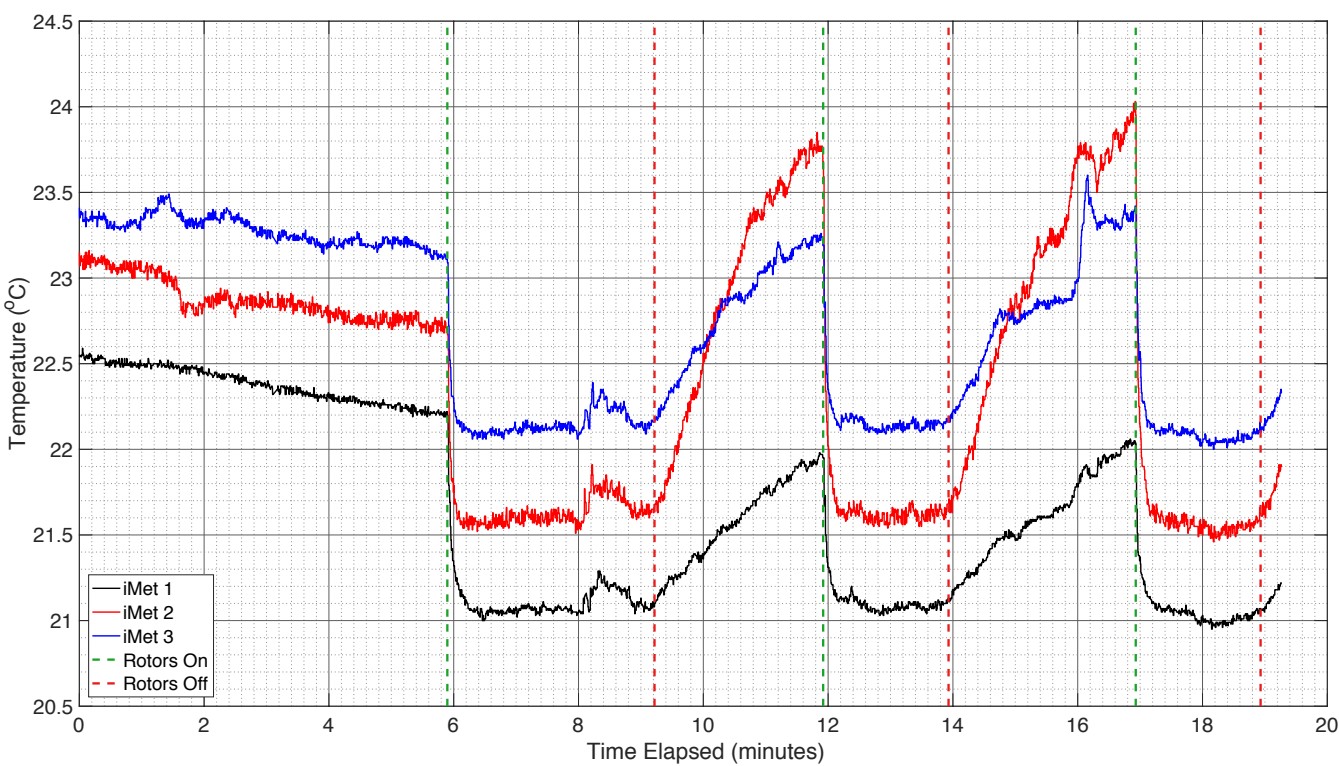

**Figure 3.** Time series of the uncalibrated iMet temperature sensors sampling at 10 Hz relative to the times the fan was switched on (green dashed line) and off (red dashed line).

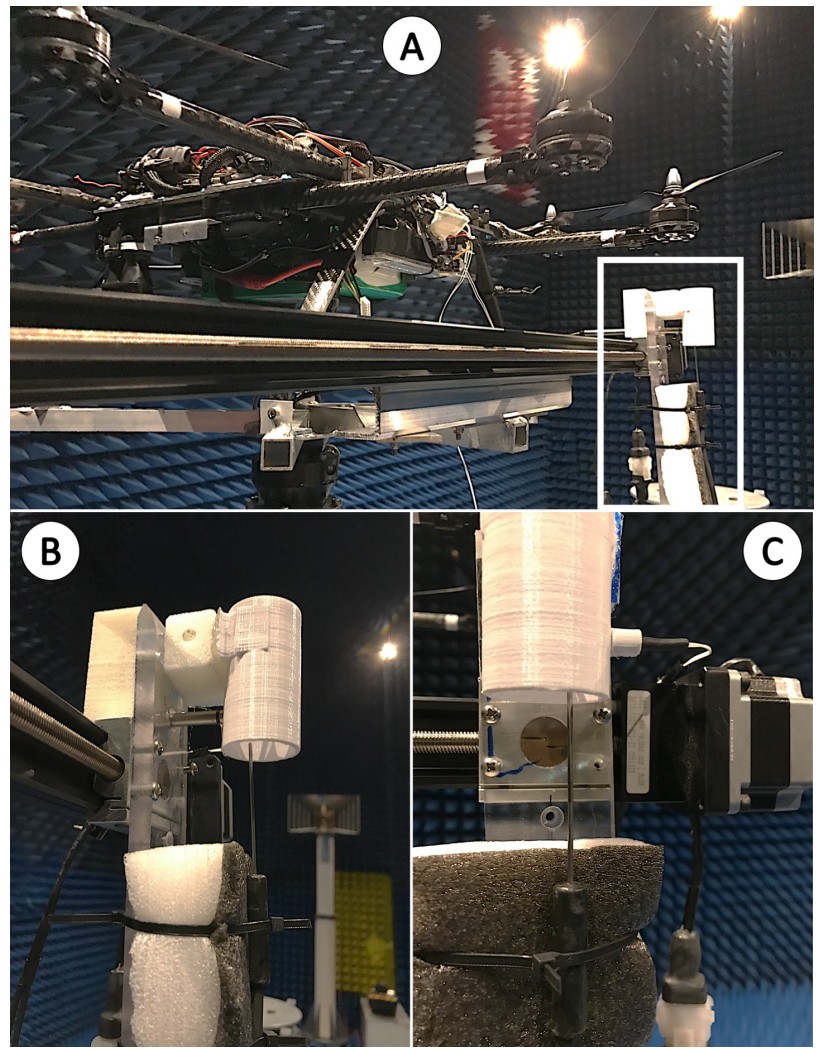

**Figure 4.** A: Position of linear actuator arm underneath rwUAS on mounting pedestal. Arm was aligned such that sensors passed directly underneath the motor mounts so as to make the system essentially two-dimensional. The sensor package is outlined in white. B: Close-up side view of the sensor package. The NSSL thermistor (CS 109) is strapped vertically to a foam mount so that it reaches inside the solar shield (white cylinder) from the bottom. The hot-wire anemometer is attached to the linear actuator arm with a clear mount and passes into a hole on the back side of the solar shield. C: Close-up front view of the sensor package. An iMet thermistor (PT 100) enters the solar shield through a hole on the right side. CS 109 also visible pointing vertically.

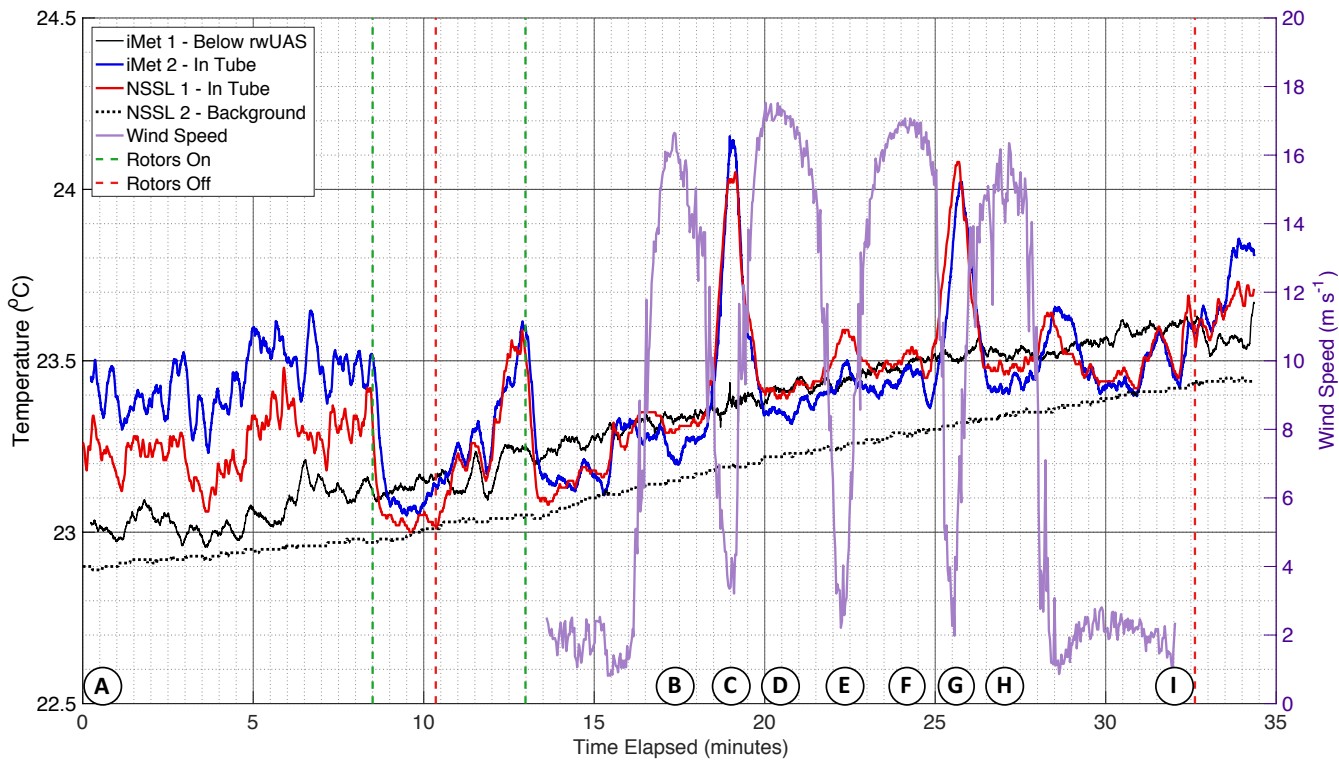

**Figure 5.** Experiment 1 - Time series graph of air temperature (°C) and wind speed (m s$^{-1}$). The background temperature is shown by the CS 109 probe (dotted black) and the iMet sensor (solid black). The CopterSonde temperatures are shown by the iMet sensor (solid blue), while the reference temperature of the CS 109 is shown in solid red. Air velocity at the CopterSonde sensor location is plotted in solid orange. Dotted green and red vertical lines indicate times when the motors were throttled on and off, respectively. Points A–I from Figure 2 are also indicated here.

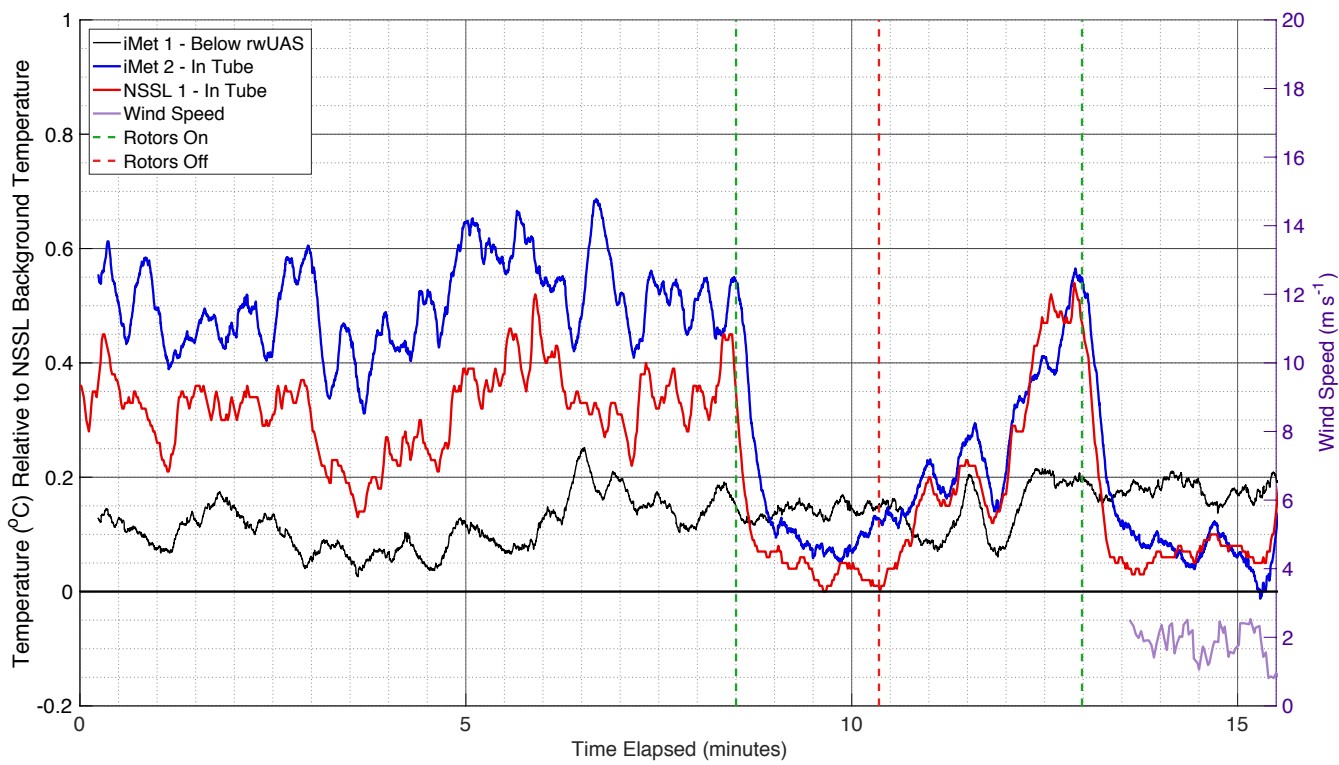

**Figure 6.** Temperatures (°C) relative to NSSL background temperature at the beginning of Experiment 1. This perspective emphasizes sensor self-heating (in addition to bias from the hot-wire anemometer) as the propellers are throttled on and off (green and red vertical dashed lines, respectively). Furthermore, the initial difference between NSSL sensors can be attributed to the presence of the anemometer in the radiation shield. While wind speeds were not recorded during this time, it is reasonable to extrapolate the 2.5 m s$^{-1}$ reading backwards from minute 13.6 since the sensor position was fixed.

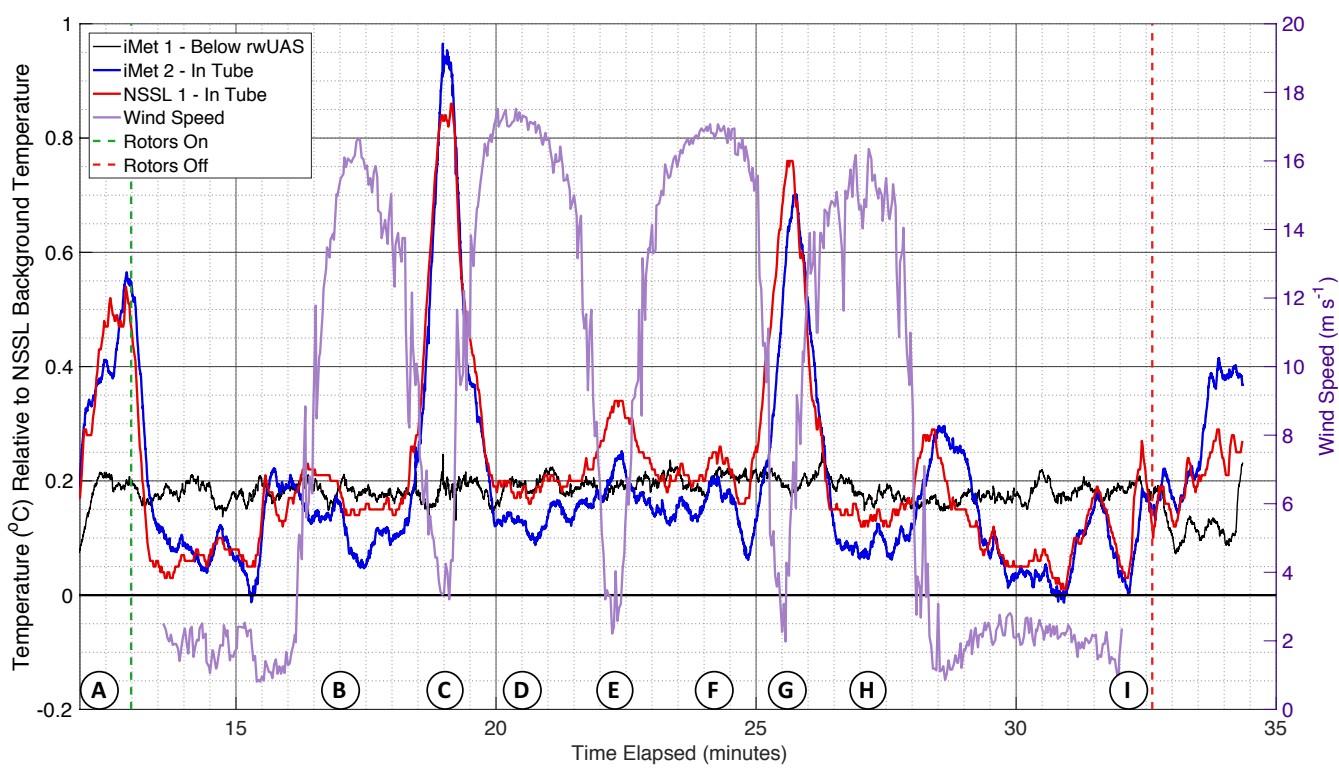

**Figure 7.** Experiment 1 time series of temperature (°C) relative to the NSSL background temperature after the actuator arm begins incrementing. Points A–I are included from Figure 2.

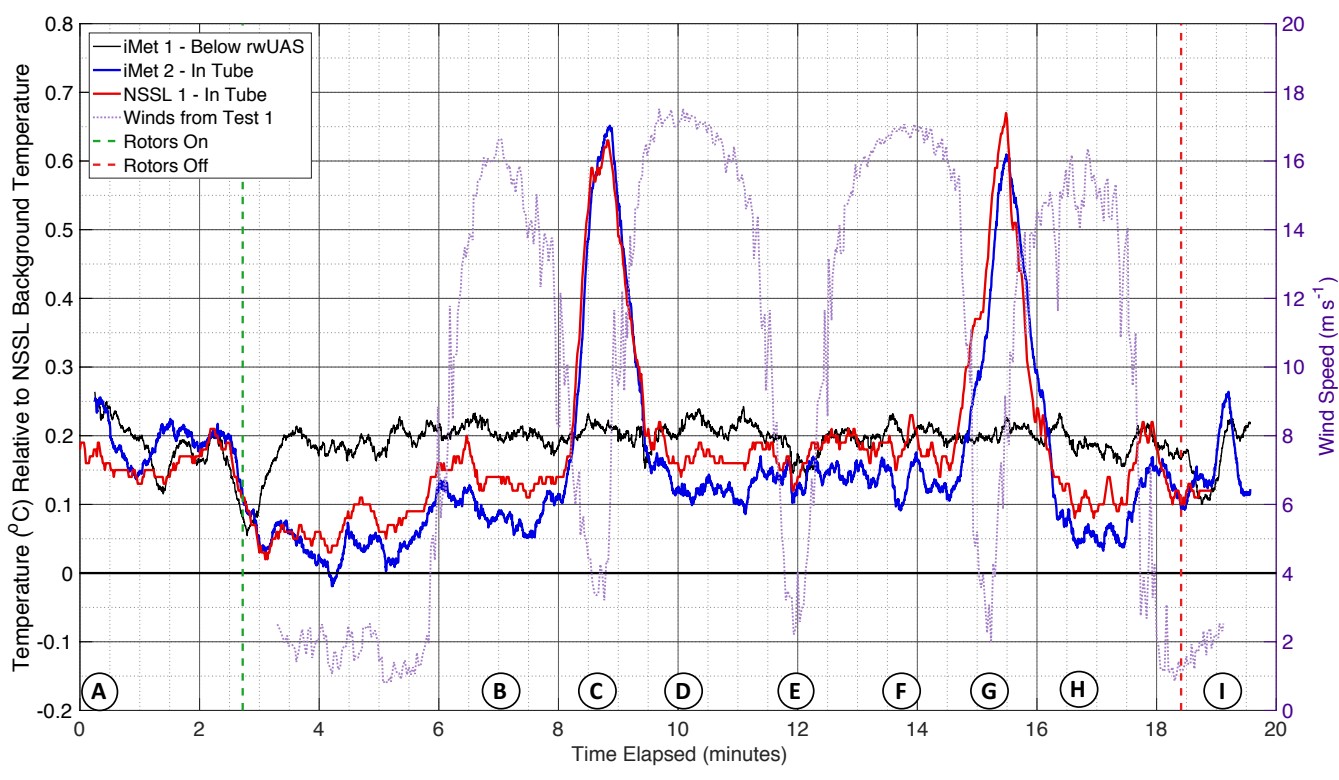

**Figure 8.** Experiment 2 - temperature and wind speed vs. relative time. Winds from Experiment 1 included for reference (dotted orange). Points A–I are included from Figure 2.

*Competing interests.* The authors declare that they have no conflicts of interest.

*Acknowledgements.* This research has been supported in part by National Science Foundation under grant number 1539070; the NOAA UAS Program Office through the Environmental Profiling and Initiation of Convection Project; and internal funding from the University of Oklahoma. The authors would like to acknowledge contributions from Arturo Umeyama for his development of the software integrating data
5  streams into the Pixhawk flight controller, as well as Bill Doyle and Brent Wolf for development of the CopterSonde and mounting bracket used in these experiments. Dr. Christopher Fiebrich also contributed to the calibration and validation efforts of the thermodynamic sensors used in this study.

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
