# Peer review of "Considerations for temperature sensor placement on rotary-wing unmanned aircraft systems"

_Atmospheric Measurement Techniques, 2018_

## Referee Comment (RC1) · Anonymous Referee #2 · 24 May 2018

General Comments: The manuscript describes comprehensive laboratory tests and observations on the optimal placement of temperature sensors on small unmanned rotary wing aircraft. This study presents the first systematic investigation on the corresponding sensor placement and is without doubt highly beneficial for the RPAS (Remotely Piloted Aircraft System) community working on atmospheric research and therefore clearly in the scope of AMT. However, at its present stage, I evaluate this manuscript in the transition between a comprehensive laboratory report and a scientific publication and suggest major revisions and improvements before considering it for final publication.

[Figure]

The introduction, in particular the first parapgraph (lines 2 to 9), is rather messy and should be restructured and in several places be formulated more accurately. It is a collection of statements that are not wrong but also not 100% correct and I miss a clear thread guiding through this motivational part. Some examples: - "The PBL is the lowest part of the troposphere which exchanges energy with the Earth's surface" would be only correct if you add "on a time scale of one hour or below"; otherwise I argue that at least the whole troposphere is exchanging energy with the surface - "balloons are costly and have limited vertical resolution", that might be correct for radiosoundings, but not for tethered balloons

Specific (major and minor) comments:

p.1, lines1-4 : the first two sentences are more motivation and not typical content of an abstract

p. 2, line 21: I suggest to replace "integration of rwUAS with observational networks" by "integration of rwUAS into observational networks"

p.3, line 1: replace ", and thus the temperature" by ", and thus the air temperature"

p. 3, line 11: " "dissipation of heat from the rotary motor": I feel that dissipation is not the correct expression here and suggest to replace "dissipation" by "emission"

p. 3, line 12: replace "Flow in proximity" by "Flow in the proximity"

p. 3, line 21: A picture of the CopterSonde, preferably outside the laboratory, would be great here to show the reader what you are talking about

p. 3, line 22: replace "total all-up weight" by "total take-off weight"

p.4, line 2: what is the underlying sensor type for the iMet temperature sensor; I assume they did not develop them from scratch on their own

p.4, line 5: replace "and is ideal" by "and are ideal"

p.4, line 26: give a proper type description/type definition for the sensor; "known as the 109" sounds rather unscientific

p.5, line 3: remove "one"

p.5, line 5: can you comment on/explain what is "as close as possible to the temperature sensor location"; a value in mm or cm would help here; in addition you should state and discuss which disturbances on your measurements you have to expect by this setup

p.5, line 21: "Prior to analysis of the sensor placement temperature profiles detailed in later sections. . ." makes in this form no sense; should it read "Prior to the analysis of the sensor placement, detailed in later sections, . . .""

p.5, line 21: "depiction" sounds wrong here; better "characterization" or "investigation"

p.5, line 29: replace "A time series of the iMet temperature responses are displayed" by "The time series of the iMet temperature response is displayed"

p.6, line 13 insert comma aroung "however"

p.8, line 20 "At minute 15.5 the probes intercepted a warm stream of air likely owing to turbulent fluctuations and compressional heating"; both figures A4 and A5 show that the temperature increase starts clearly (about 30 s) before the increase in the wind speed, therefore I doubt in the explanation of compressional heating

p.8, general please add arrows and labels to the plots showing where exactly you want the reader to look at and refer to those labels in the text (in addition to the time stamp you give); see also comment on figure 4A

p.8, line 33 "the primary driver of this temperature rise was likely a warm air stream originating from the hot wire anemometer"; couldn't it also be simply self-heating?

p.9, line 10 insert "level" after "critical"

p.9, line 17 ".. as other trials not described. . ."; as the manuscript is not so long, you might consider to include some additional material in it

comments on figures: • Is there any reason for numbering the Figures with A1, A2, etc instaed of simply using 1, 2, . . . • I suggest to remove the figure titles, e.g. "Coptersonde Fan Aspiration Experiment" in 2,4,5, and 6; this is redundant information to the figure caption • The figures of the time series are not really consistent in their layout

figure A2: Have you an explanation for the "bump" in all three temperature time series around 8 minutes? Could this be an effect of a circulation that builds up inside the chamber? I have seen similar structures in flow test experiments with quadcopters in a larger hall; a pity that you shortened the the motor-on time to 2 minutes (I hope this did not happen on purpose to avoid a discussion of this issue); the legend could be shortened (iMet 1, iMet 2,. . ..), it is clear that it is temperature here

figure A3: the two pictures are not very illustrative, mainly due to the low contrast; it is highly recommended to replace them by better ones

figure A4: "Compressional heating off tip of propeller at minute 15.5 just as wind speeds picks up"; what I see is; wind picks up at 16, while the main temperature jump happens at 15:30; how can this be related to compressional heating? I also highly recommend to include markers e.g. "A", "B", "C" and so on, in the figure to exactly point out for the reader what specific feature you are describing in the text; this was really hard to figure out without this help; What is the reason to show the temperatures here as absolute values, when they will be presented in all follow-on figures as differences relative to the NSSL background?

figures A6: for a better comparison with experiment 1 you should also present the second half of figure A4 in the form relative to the NSSL background temperature

---

## Referee Comment (RC2) · Anonymous Referee #3 · 28 Jun 2018

This is an interesting paper which describes measurements with temperature sensors at differenet locations under the rotors of an octocopter drone. The measurements are made in the laboratory, so the issues with aerodynamic stability of the aircraft, that follow on from this work, are not addressed. In summary I think that subject to some revision to improve the clarity (see the points below) this paper should be accepted for publication in AMT.

1) Why thermistors, and not fast thermocouples which have fewer issues from self heating?

2) page 4, line 5 "is" should be "are"

[Figure]

3) What is the solar shield made of? In particular is it opaque to solar IR radiation - many plastics are not. (I realise that this is irrelevant to the laboratory measurements, but presumably the same shield is intended for use in the field.)

4) The description of the various sensors is a bit confusing. One reads that the sensors are iMet thermistors in section 2.2. Then one reads in the next section that the sensors are Campbell Scientific ones - and in "the sensors" I emphasise "the". (As an aside the sentence with " ... two stainless steel probes from Campbell Scientific known as the 109 were used." is an odd way of putting it. Are they model 109 or what?) I think that the different sensors are there for intercomparison but it isn't clear from the way that section 2 is written, and it wouldn't hurt to be a bit clearer and more explicit at this early stage of the paper.

5) The caption of Figure A3 needs work; b) and c) need to be better described to make it clear what the different things that they show are. I can work out what is shown in b) by looking at a) but I can't be sure about c). Is c) the same thing from a different angle? Would a line drawing be more effective here? Certainly it would help to have a circle drawn on the photo on a) to indicate where the detail illustrated in b) is located. It would also be good if the main photo could be retaken with better attention to lighting as it is not a very clear photo. The ceiling light behind one of the rotors is very distracting and degrades the contrast of the apparatus that is the main subject of the picture.

---

## Author Comment (AC1) · 25 Jul 2018

The authors would like to thank the reviewer for their insightful questions and feedback. Included in the supplement are responses to their comments and a revised version of the manuscript detailing changes.

Please also note the supplement to this comment: https://www.atmos-meas-tech-discuss.net/amt-2018-65/amt-2018-65-AC1-supplement.zip

---

## Author Response (AR1)

The authors would like to thank the reviewers and editors for their insightful questions and feedback. These comments have undoubtedly improved the quality of this manuscript. Author responses to each individual comment are outlined below.

**AMTD Author's Response to Anonymous Referee #2**
Referee's comments are **bold and italicized**, and authors' are plain text.

***General Comments: The manuscript describes comprehensive laboratory tests and observations on the optimal placement of temperature sensors on small unmanned rotary wing aircraft. This study presents the first systematic investigation on the corresponding sensor placement and is without doubt highly beneficial for the RPAS (Remotely Piloted Aircraft System) community working on atmospheric research and therefore clearly in the scope of AMT. However, at its present stage, I evaluate this manuscript in the transition between a comprehensive laboratory report and a scientific publication and suggest major revisions and improvements before considering it for final publication. The introduction, in particular the first parapgraph (lines 2 to 9), is rather messy and should be restructured and in several places be formulated more accurately. It is a collection of statements that are not wrong but also not 100% correct and I miss a clear thread guiding through this motivational part. Some examples: - "The PBL is the lowest part of the troposphere which exchanges energy with the Earth's surface" would be only correct if you add "on a time scale of one hour or below"; otherwise I argue that at least the whole troposphere is exchanging energy with the surface - "balloons are costly and have limited vertical resolution", that might be correct for radiosoundings, but not for tethered balloons.***

The introduction and literature review has been restructured and expanded upon for clarity and emphasis. The first four paragraphs now read:

> The planetary boundary layer (PBL) is the lowest layer of the troposphere which exchanges energy with the Earth's surface on timescales of less than one hour (Stull, 1988), and acquiring atmospheric measurements in this region has proven to be challenging (National Research Council, 2009; Hardesty and Hoff, 2012). PBL flows are highly complex and nonlinear in space and time, even with several layers of assumptions applied in theory. As such, it has always been a challenge for atmospheric scientists to collect representative measurements of the environment, even with continual advances in technology. One of the most common resources for PBL studies has been instrumented towers, which can continually provide data at a point location over long periods of time (e.g., Charba, 1974; Shapiro, 1984; Poulos et al., 2002). While highly reliable and configurable, instrumented towers do come with an inherent downside. Being limited in vertical extent, the convective boundary layer often extends well above even the tallest of towers. Even networks with the 30 km average horizontal resolution of the Oklahoma Mesonet (Brock et al., 1995; McPherson et al., 2007) still cannot provide details on the vertical structure of the atmosphere.
>
> Presently, weather balloons are the most common tool available for in-situ observations above the level of towers. They provide valuable kinematic and thermodynamic data from the upper atmosphere, which impacts both short-term weather forecasts (Cohen et al., 2007; Faccani et al., 2009; Lackmann, 2011) as well as

climatological trends (Luers and Eskridge, 1998; Lanzante et al., 2003; Thompson and Solomon, 2005) and can serve as a baseline for model verification (Agustí-Panareda et al., 2010; Benjamin et al., 2010; Gensini et al., 2014). Rawinsondes are launched in hundreds of locations around the world every day, although usually only twice a day at most sites. This operational network is also not suited to provide adequate PBL measurements, as they ascend too rapidly through the lowest levels (National Research Council, 2009). More frequent deployments with slower ascents are commonly performed on field campaigns (e.g., Kosiba et al., 2013; Parker, 2014; Trapp et al., 2016; Geerts et al., 2017), but this becomes expensive as the sensor package is rarely recovered for reuse. Specialized satellite remote sensors can derive vertical thermodynamic and kinematic profiles across significant areas of the Earth, but vertical resolutions in the PBL are too coarse for practical application.

Surface-based remote sensors such as wind profilers, Doppler lidars, sodars, and radiometers are capable of continuously observing a fixed location (e.g., Grund et al., 2001; Poulos et al., 2002; Banta et al., 2015; Bonin et al., 2015; Lundquist et al., 2017; Toms et al., 2017; Geerts et al., 2017; Blumberg et al., 2017), but rely on numerous assumptions about the atmosphere and have trouble resolving measurements close to the surface. These types of instruments are also cost-prohibitive when considering expansion to larger-scale networks such as the global upper-air sites.

Even when combining surface towers, balloons, and remote sensors with other observational techniques such as tethered balloons, Doppler weather radars, and satellite remote sensors, the National Research Council (2009) still concluded that the "vertical component of U.S. mesoscale observations is inadequate." The NRC in this report implored government agencies to pursue developments in capabilities to monitor the lower atmosphere at finer scales in space and time.

***p.1, lines1-4 : the first two sentences are more motivation and not typical content of an abstract***

Sentences rephrased. Beginning now reads:

Integrating sensors with a rotary-wing unmanned aircraft system (rwUAS) can introduce several sources of biases and uncertainties if not properly accounted for. To maximize the potential for rwUAS to provide reliable observations, it is imperative to have an understanding of their strengths and limitations under varying environmental conditions. This study focuses on the quality of measurements relative to sensor locations on board rwUAS.

***p. 2, line 21: I suggest to replace "integration of rwUAS with observational networks" by "integration of rwUAS into observational networks"***

Fixed

***p.3, line 1: replace ", and thus the temperature" by ", and thus the air temperature"***

Fixed

*p. 3, line 11: " "dissipation of heat from the rotary motor": I feel that dissipation is not the correct expression here and suggest to replace "dissipation" by "emission"*

Agreed, emission seems more appropriate in this situation

*p. 3, line 12: replace "Flow in proximity" by "Flow in the proximity"*
Fixed

*p. 3, line 21: A picture of the CopterSonde, preferably outside the laboratory, would be great here to show the reader what you are talking about*

Figure added

*p. 3, line 22: replace "total all-up weight" by "total take-off weight"*

We prefer the term "all-up". In aerospace engineering, take-off weight is used when aircraft weight changes in flight (e.g., fuel burning), but since the UAV's weight is unchanged, all-up weight is more accurate.

*p.4, line 2: what is the underlying sensor type for the iMet temperature sensor; I assume they did not develop them from scratch on their own*

Changed to "PT 100 thermistors distributed by International Met Systems (iMet)"

*p.4, line 5: replace "and is ideal" by "and are ideal"*

Fixed

*p.4, line 26: give a proper type description/type definition for the sensor; "known as the 109" sounds rather unscientific*

Changed to "two Campbell Scientific model 109 thermistors (CS 109) were used"

*p.5, line 3: remove "one"*

Changed to "Doing so can indicate the extent…"

*p.5, line 5: can you comment on/explain what is "as close as possible to the temperature sensor location"; a value in mm or cm would help here; in addition you should state and discuss which disturbances on your measurements you have to expect by this setup*

Changed to "about 1 cm" and added "On these scales, special considerations regarding the anemometer as a heat source were also required. As will be discussed in Section 4, a separate trial to control for possible interference was also conducted."

*p.5, line 21: "Prior to analysis of the sensor placement temperature profiles detailed in later sections. . ." makes in this form no sense; should it read "Prior to the analysis of the sensor placement, detailed in later sections, . . .""*

Changed to "Prior to analysis of the sensor placements (detailed in Sections 4 and 5),…"

*p.5, line 21: "depiction" sounds wrong here; better "characterization" or "investigation"*
Changed to "characterization"

*p.5, line 29: replace "A time series of the iMet temperature responses are displayed" by "The time series of the iMet temperature response is displayed"*

These lines discussing the figure were moved to the caption and replaced by: "Analysis of the iMet temperature responses (Figure 3) revealed that the sensors closely tracked with one another while the fan was on, with only a linear offset."

*p.6, line 13 insert comma aroung "however"*

Fixed

*p.8, line 20 "At minute 15.5 the probes intercepted a warm stream of air likely owing to turbulent fluctuations and compressional heating"; both figures A4 and A5 show that the temperature increase starts clearly (about 30 s) before the increase in the wind speed, therefore I doubt in the explanation of compressional heating*

We believe that this warm air stream associated with turbulent fluctuations and compressional heating spreads downwards on the periphery of the primary propeller wash. This magnitude of heating seems reasonable when coupled with this explanation, as the propellers rotate close to 10,000 RPM. Air interacting with the tips will heat the most, and mix turbulently with environmental air if not forced directly downwards. More description was added to emphasize that the warm air is along the extremity of the propeller wash. It now reads: "…spreading down and outward along the periphery of the propeller wash."

*p.8, general please add arrows and labels to the plots showing where exactly you want the reader to look at and refer to those labels in the text (in addition to the time stamp you give); see also comment on figure 4A*

Added points A-I on figures for reference, inserted throughout text for clarity.

*p.8, line 33 "the primary driver of this temperature rise was likely a warm air stream originating from the hot wire anemometer"; couldn't it also be simply self-heating?*

Yes, that is certainly possible. Changed to say it is likely a combination of self-heating and hot-wire influence.

*p.9, line 10 insert "level" after "critical"*

Fixed

**p.9, line 17 ".. as other trials not described. . ."; as the manuscript is not so long, you might consider to include some additional material in it**

We ran similar but less well-organized experiments that all elude to similar results. New discussion has been added:

> "In addition to the two experiments discussed previously, several test trials were also conducted. These additional experiments were almost identical to the setup described in Section \ref{sec:chamber}, but were missing some key elements. For instance, the sensors were mounted to the linear actuator arm without a solar radiation shield. As a result, the sensors intercepted larger areas of influence from heat sources such as the motors and propeller tips when compared to sensors inside a shield. Furthermore, the environmental air was not mixed using a carpet fan prior to running the experiment. There was a much more noticeable increase in the chamber's background temperature during these trials, which reduced confidence in analysis and was difficult to reproduce. These tests ultimately still eluded to similar results in temperature sensor placement, prompting the more refined experiments as the focus of this study."

**comments on figures: âAˇ c Is there any reason for numbering the Figures with A1, ´ A2, etc instaed of simply using 1, 2, . . . âAˇ c I suggest to remove the figure titles, e.g. ´ "Coptersonde Fan Aspiration Experiment" in 2,4,5, and 6; this is redundant information to the figure caption âAˇ c The figures of the time series are not really consistent in their ´ layout**

Removed the appendix and "A" from figure numbers as well as figure titles to mitigate redundancy.

**figure A2: Have you an explanation for the "bump" in all three temperature time series around 8 minutes? Could this be an effect of a circulation that builds up inside the chamber? I have seen similar structures in flow test experiments with quadcopters in a larger hall; a pity that you shortened the the motor-on time to 2 minutes (I hope this did not happen on purpose to avoid a discussion of this issue); the legend could be shortened (iMet 1, iMet 2,. . ..), it is clear that it is temperature here**

This temperature "bump" is due to someone walking in front of the ducted fan to take a picture during the experiment; it was only after post processing that we realized this was actually a measureable phenomenon. The motor-on time of two minutes was not chosen for any particular reason aside from experience in the field telling us that these sensors usually react to aspiration in a matter of seconds, and two minutes is a reasonable amount of time in flight when performing vertical profiles up to 120 m AGL (the limit imposed by the FAA in the US in unrestricted airspace). This explanation has been incorporated.

**figure A3: the two pictures are not very illustrative, mainly due to the low contrast; it is highly recommended to replace them by better ones**

Unfortunately, it is not possible to acquire new photos of this setup. In its place, higher quality versions of the original photos with some digital enhancement have been substituted. We do agree that the low quality photos are distracting, and ideally these new ones provide more clarity to the reader about the physical system in this experiment. We think that Figure 2 provides enough of the technical details missing in these pictures.

*figure A4: "Compressional heating off tip of propeller at minute 15.5 just as wind speeds picks up"; what I see is; wind picks up at 16, while the main temperature jump happens at 15:30; how can this be related to compressional heating? I also highly recommend to include markers e.g. "A", "B", "C" and so on, in the figure to exactly point out for the reader what specific feature you are describing in the text; this was really hard to figure out without this help; What is the reason to show the temperatures here as absolute values, when they will be presented in all follow-on figures as differences relative to the NSSL background?*

Caption updated to remove scientific discussion. Markers A-I included for clarity. New figure of Experiment 1 with relative temperature time series now included. See earlier comment for discussion on compressional heating.

*figures A6: for a better comparison with experiment 1 you should also present the second half of figure A4 in the form relative to the NSSL background temperature*

Added as Figure 7

**AMTD Author's Response to Anonymous Referee #3**
Referee's comments are ***bold and italicized***, and authors' are plain text.'

*This is an interesting paper which describes measurements with temperature sensors at differenet locations under the rotors of an octocopter drone. The measurements are made in the laboratory, so the issues with aerodynamic stability of the aircraft, that follow on from this work, are not addressed. In summary I think that subject to some revision to improve the clarity (see the points below) this paper should be accepted for publication in AMT.*

*1) Why thermistors, and not fast thermocouples which have fewer issues from self heating?*

In general, thermistors tend to be more sensitive to small changes in temperature when compared to thermocouples. This feature is advantageous when probing small scales of the atmosphere with small temperature ranges. Otherwise, thermocouples do have the advantage in that they are more linear in their calibration and less self-heating, and would be worth investigating in future studies with UASs.

*2) page 4, line 5 "is" should be "are"*

Fixed

*3) What is the solar shield made of? In particular is it opaque to solar IR radiation - many plastics are not. (I realise that this is irrelevant to the laboratory measurements, but presumably the same shield is intended for use in the field.)*

The solar shields are 3D printed using PLA filament, which is typically transparent to IR radiation. That is something worth considering for future designs, as these were the same ones as used in field applications. Thank you for the consideration of this.

*4) The description of the various sensors is a bit confusing. One reads that the sensors are iMet thermistors in section 2.2. Then one reads in the next section that the sensors are Campbell Scientific ones - and in "the sensors" I emphasise "the". (As an aside the sentence with " ... two stainless steel probes from Campbell Scientific known as the 109 were used." is an odd way of putting it. Are they model 109 or what?) I think that the different sensors are there for intercomparison but it isn't clear from the way that section 2 is written, and it wouldn't hurt to be a bit clearer and more explicit at this early stage of the paper.*

More details about the individual sensors were added, and the confusing colloquialisms were removed. The iMet sensors are PT 100 thermistors, and the NSSL sensor is a Campbell Scientific model 109 thermistor (CS 109). The rest of the paper was changed to refer to the NSSL sensors as the CS 109.

*5) The caption of Figure A3 needs work; b) and c) need to be better described to make it clear what the different things that they show are. I can work out what is shown in b) by looking at a) but I can't be sure about c). Is c) the same thing from a different angle? Would a line drawing be more effective here? Certainly it would help to have a circle drawn on the photo on*

***a) to indicate where the detail illustrated in b) is located. It would also be good if the main photo could be retaken with better attention to lighting as it is not a very clear photo. The ceiling light behind one of the rotors is very distracting and degrades the contrast of the apparatus that is the main subject of the picture.***

Unfortunately, it is not possible to acquire new photos of this setup. In its place, higher quality versions of the original photos with some digital enhancement have been substituted. We do agree that the low quality photos are distracting, and ideally these new ones provide more clarity to the reader about the physical system in this experiment. We think that Figure 2 provides enough of the technical details missing in these pictures.

The captions were updated to provide more visual descriptions, and an outline was included to clarify relation of images B and C to A.

[revised manuscript text omitted]

---

## Author Response (AR2)

The authors are grateful for the reviewers and editors for their revisions and feedback, which have greatly enhanced the effectiveness of this manuscript. Author responses to each individual comment are outlined below.

**AMTD Author's Response to Associate Editor Minor Revisions**
Associate editor's comments are ***bold and italicized***, and authors' are plain text.

**1)** ***The expression "subtract off of" is a colloquialism, and should be replaced with "subtract from", or "subtract off" if the direct object is the part that is removed.***

The expression "subtract off of" does not appear in the text of this manuscript. No change made.

**2)** ***The referee commented on the use of the word "dissipation" and suggested replacing it by "emission". However that implies (or could imply) only radiative heating of the sensor. I'm pretty sure that you had advection of heat in mind. I think that "Furthermore, heat from the rotary motor …" is simpler and more general.***

We agree that the word "emission" could imply radiative heating, which is too specific. Text changed to read "Furthermore, heat from the rotary motor…"

**3)** ***In the some of the figures which show graphs (especially figure 8), you have used orange lines and text. This doesn't show up all that well on my screen, and if a reader chooses to print in black and white, it won't show up at all. This colour should be changed to a darker one that contrasts better with the white background.***

This color has been changed to purple in Figures 5 – 8. Additionally, red and blue intensity of iMet 2 and NSSL 1 have been reduced slightly. Line width of iMet 1 was also reduced.

[revised manuscript text omitted]